# A Prospective, Longitudinal Evaluation of SARS-CoV-2 COVID-19 Exposure, Use of Protective Equipment and Social Distancing in a Group of Community Physicians

**DOI:** 10.3390/healthcare10020285

**Published:** 2022-02-01

**Authors:** Eli D. Ehrenpreis, Sigrun Hallmeyer, David H. Kruchko, Alexea A. Resner, Nhan Dang, Natasha Shah, Nancy Mayer, Anne Rivelli

**Affiliations:** 1Department of Medicine, Advocate Lutheran General Hospital, Park Ridge, IL 60068, USA; Sigrun.Hallmeyer@aah.org (S.H.); David.Kruchko@aah.org (D.H.K.); lexresn2@yahoo.com (A.A.R.); ndang3188@gmail.com (N.D.); shah.natasha49@gmail.com (N.S.); Anne.Rivelli@aah.org (A.R.); 2Rosalind Franklin University of Medicine and Science, North Chicago, IL 60064, USA; 3E2Bio Life Sciences, LLC, Evanston, IL 60201, USA; 4Inspira Medical Center, Vineland, NJ 08360, USA; 5Chicago College of Osteopathic Medicine, Midwestern University, Downers Grove, IL 60515, USA; 6Chicago Research Center, Inc., Chicago, IL 60634, USA; nmayer43@midwestern.edu

**Keywords:** coronavirus, COVID-19, SARS-CoV-2, pandemic, epidemic, antibodies, IgG, personal protective equipment, PPE, social distancing, epidemiology, immunity, McNemar test, healthcare workers, physician

## Abstract

Introduction: Healthcare workers experience a significant risk of exposure to and infection from SARS-CoV-2, COVID-19. Nonetheless, little research has focused on physicians’ use of personal protective equipment (PPE), their concerns about becoming infected and their social distancing maneuvers. Methods: All staff physicians at Advocate Lutheran General Hospital were invited to participate. Their COVID-19 IgG antibody level was measured and an online questionnaire was completed. The questionnaire assessed the risk of COVID-19 exposure, PPE usage, concern for contracting COVID-19, the performance of high-risk procedures, work in high-risk settings, and social distancing practices. Testing was performed in September (T0), and December 2020 (T1) at the height of the global pandemic. Results: A total of 481 (26.7%) of 1800 AGLH physicians were enrolled at T0 and 458 (95% of the original group) at T1. A total of 21 (4.3%) and 39 (8.5%) participants had antibodies at T0 and T1. A total of 63 (13.8%) worked in high-risk settings and 111 (24.2%) performed high-risk procedures. Participants working in high-risk settings had increased exposure to COVID-19 infected patients (OR = 4.464 CI = 2.522–8.459, *p* < 0.001). Participants were highly adherent to the use of PPE and social distancing practices including mask-wearing in public (86%, 82.1%), avoiding crowds (85.1%, 85.6%), six feet distancing (83.8%, 83.4%), and avoiding public transportation (78%, 83.8%). A total of 251 (55.4%) participants expressed moderate to extreme concern about becoming infected with COVID-19. Conclusions and Relevance: Among a group of community physicians, consistent PPE use and social distancing practices were common. These practices were associated with a low level of initial acquisition of COVID-19 infections and a relatively low longitudinal risk of infection.

## 1. Introduction

The severe acute respiratory syndrome coronavirus-2 (SARS-CoV-2), COVID-19 pandemic has redefined the focus of the global medical community due to its high infection rate and significant morbidity and mortality. As of 30 August 2021, the documented case-fatality rate of COVID-19 was 1.6% in the United States [1]. Multiple factors including socioeconomic status, the presence of significant comorbidities, and patient demographics all influence the infectivity of the virus and the severity of infection [2,3,4]. Due to the requirements of their professions, healthcare workers have been the frontline in the management of patients with COVID-19 infections. Physicians and other healthcare providers are required to have an intensified commitment that includes the use of protective measures for themselves and their colleagues. They have been faced with the potential sacrifice of their own health and lives in the process. Healthcare workers were classified as a high-risk group due to contact with infected patients [5,6]. Physicians involved in the performance of high-risk procedures (such as intubation and bronchoscopy) are likely at a higher risk for direct exposure to the virus [7]. Furthermore, healthcare workers in high-risk settings, such as the emergency department (ED) and intensive care units (ICU), may be more likely to acquire the infection [8,9,10]. Involvement in high-risk procedures also occurs in these settings. When treating patients with COVID-19, healthcare workers are instructed to use personal protective equipment (PPE) such as N95 masks, face shields, gowns, and gloves. Initially, the availability of PPE was a concern for healthcare workers across the country. As the pandemic progressed, PPE became more readily available nationwide [11]. It has generally been assumed that the higher risk of COVID-19 infection in healthcare workers was mitigated, to some extent, by using PPE [12]. According to guidelines from the Center for Disease Control (CDC), the primary protection against community exposure to COVID-19 infection is social distancing measures and the wearing of a well-fitting mask while in public [13]. In general, these measures appear to be effective in reducing the spread of the disease and possibly provide individual risk reduction [14].

Data suggest that exposure to COVID-19 infections may be documented with the measurement of serum anti-COVID-19 immunoglobulin G (IgG) antibody levels [15,16,17]. Unlike direct measurements for the presence of COVID-19 viral particles using polymerase chain reaction (PCR), IgG antibody testing can determine previous infections, whether symptoms were present or not. IgG antibody testing also may have a role in determining the likelihood of developing COVID-19 infection in persons previously infected with or vaccinated against COVID-19 [18]. COVID-19 IgG antibodies become detectable in the serum within 2–3 weeks after illness onset, though data on how long they remain detectable in serum are still limited [19]. Until the advent of the Omicron variant, reinfection with COVID-19 in vaccinated individuals was relatively rare. These were previously attributed to the new Delta and Lambda variants of the virus [20].

Abbott Laboratories has developed and distributed an IgG test to determine if an individual has virus-specific IgG antibodies against COVID-19. This test has a reported sensitivity of 100% and specificity of 99.9% if performed at least 14 days after becoming infected with COVID-19 [21,22]. Currently, this test is used at our center in patients with known and recovered COVID-19 infections who were asymptomatic for at least 2 weeks to document adequate immunologic response and for research purposes.

Despite the evolution of the COVID-19 pandemic in the United States and other countries to the post-vaccination era, the approach by healthcare systems during the initial phases of COVID-19 requires further examination. To the present, there is little information on physicians’ usage of PPE. Prior studies on PPE use have focused on high-risk medical providers including dentists [23] and otolaryngologists [24]. These studies have shown general compliance with the use of PPE when working directly with patients. Yet these studies provide little understanding of protective measures utilized by practitioners outside of their work environment. In addition, these studies have focused on specific, high-risk specialists and have not provided an overview that encompasses PPE usage in physicians with a variety of specialties working in a hospital setting. This information will provide a more direct real-world understanding of PPE usage among physicians. Furthermore, there is little longitudinal information on attitudes and safety practices among practicing physicians. This is an important area of focus since prior studies have failed to determine whether practices have changed with surges of the pandemic, or whether factors such as social distancing fatigue effects community physicians as a group. In addition, physician attitudes regarding their own risk of COVID-19 and possibly exposing their families to COVID-19 during the pandemic have not been formally described. These data, in conjunction with the development of a new method to evaluate these issues, provide a framework for understanding the impact of the COVID-19 on physicians at the community level. Furthermore, this study and its results can function as a template for analysis and intervention to mitigate the negative effects of the pandemic on global physician practices.

In the present study, a longitudinal, survey-based investigation was performed on approximately 500 physicians at a large community-based hospital outside of Chicago, Illinois beginning in September of 2020. Ongoing COVID-19 exposure in the group was assessed by measurement of serum COVID-19 IgG antibodies. For this study, data collected from initial questionnaires were collected in September 2020 (Time 0 or T0), and follow-up questionnaires and serum sampling took place in December 2020 (Time 1 or T1) during regional surges in COVID-19 cases.

## 2. Methods

This study was created during the initial phases of the COVID-19 pandemic when the investigators identified a need to rapidly collect data related to concern about the risk of becoming infected with the virus, use of PPE and social distancing practices among the practicing physicians in our institution. At that time and to the present, no validated questionnaires have been created and are available to study the issues that were investigated in our study. Formal validation of our questionnaire is anticipated in future analyses of these data. This questionnaire may then be utilized as an investigative tool that can be utilized by other investigators. The initial version of the questionnaire was created by two researchers that were experienced with prior publications based on questionnaires that they developed (EDE and NS). Face validity testing of the questionnaire was performed by a committee of nine practitioners in the Advocate Lutheran General system that were planning to volunteer for the study. Committee members evaluated the entire questionnaire and rated each question on its appropriateness and reliability for the specific issue being questioned. Based on these ratings, a second version of the questionnaire was created and reviewed by the committee. A series of online meetings were held that resulted in the final version of the questionnaire. A complete list of all physicians on staff at Advocate Lutheran General Hospital was provided to the investigators by the medical staff office. All staff members have an institutional email. Invitations were sent to all physicians, with follow-up emails sent every 5 days three times to those that did not respond to the initial emails. Participants signed up for the study when replying to the email and signed their consent form when having their antibody tests performed. Selection bias was avoided by inviting all 1800 physician staff members. A subjective estimated minimal sample size of 25% of the entire group of physicians was anticipated by the investigators. Prior to the initiation of the study, a magnitude of effect was estimated to be represented by a greater than 10% real-world difference between groups.

Questionnaire-based data were collected as a component of a study to measure the presence of COVID-19 IgG antibodies in physicians in the Advocate Lutheran General Hospital (ALGH) hospital system. The study was initiated on 1 August 2020. Attending physicians, fellows, and residents working on staff from at least 1 March 2020, were invited to participate. Introductions of the study were sent to all on-staff providers’ work email addresses. Those who expressed interest were given additional study information, the informed consent for review, a link to sign up for a time slot for consent and blood draw, and a SARS-CoV-2 IgG assay fact sheet provided by Abbott Laboratories, Inc. The initial enrollment event took place in the hospital over 10 days between 14 September 2020 and 25 September 2020. Resident physicians and research coordinators on the study followed GCP guidelines and obtained informed consent from the participants during their appointment. Blood for one 3.5 mL serum-separating tube was collected from each participant. Serum SARS-CoV-2 IgG antibody testing was performed using chemiluminescent microparticle immunoassay (CMIA) technology. The assay used measured the presence of SARS-CoV-2 nucleocapsid protein antibodies that are produced from a previous COVID-19 infection. This test received Emergency Use Authorization (EUA) for COVID-19 antibody testing from the Food and Drug Administration (FDA) [17,21]. Quantitative results are reported as positive for index values greater than or equal to 1.4 S/C for N protein IgG and are reported negative for index values less than 1.4 S/C. The index value is calculated by dividing the sample result by the stored calibrator result (S/C). The assay is both highly sensitive and specific for COVID-19 antibodies with a 98.7% sensitivity if greater than 14 days post-exposure. The specificity for the assay (negative result agreement) for specimens collected pre-COVID-19 outbreak is 99.2% [22].

To take part in the study, participants agreed to a total of four blood draws and to answer four questionnaires over the course of 1 year: the first at the time of enrollment, and again at 3, 6, and 12 months after enrollment. The questionnaires were created as surveys in the REDCap data collection platform (REDCap, Vanderbilt University, Nashville, TN). An individualized link was sent by email to participants after they received their blood draw for survey completion. Sixty-six items were included in the first questionnaire (baseline/T0) composed of, but not limited to; demographics, relevant medical history, job information, travel history, and household information. Minor modifications are allowed per protocol for upcoming versions of the questionnaires to cover new relevant information. All four questionnaires are planned to include questions about contact history with patients infected with COVID-19, procedural risk history, personal COVID history, PPE use, social distancing measures, attitude towards social distancing, concern for contracting COVID-19, and willingness to donate convalescent plasma if positive for COVID-19 antibodies. Procedures defined as high-risk included bag valve mask ventilation, oropharyngeal suctioning, endotracheal intubation, nebulizer treatment, cardiopulmonary resuscitation, bronchoscopy, upper endoscopy, tracheostomy, and “other” as defined by the participant.

Those who tested positively received notification with their test result email to invite up to two household members to also receive COVID-19 antibody testing. Inclusion criteria for household members included being over the age of 18 and had to have been living with the physician participant for at least 2 weeks since 1 March 2020. As a part of their participation, household members also agreed to complete questionnaires. This process of physician testing, questionnaire completion, resulting, and household member invitation and testing was repeated at 3 months (December 2020) (T1) and with additional testing to be carried out at 6 months (March 2021) (T2) and 12 months (September 2021) (T3). Statistical analyses were performed by calculating frequencies and using McNemar’s test to compare T0 and T1 data as part of an interim analysis. For example, the consistency of exposure and use of high-risk aerosolizing procedures were compared from T0 to T1 and assessed for statistical significance. This study was approved by the Advocate Aurora Institutional Review Board.

The study is registered on the website ClinicalTrials.gov Identifier: NCT04540484.

### Statistical Analysis

Normally distributed data were presented as mean (SD), non-normally distributed data as median (IQR), and categorical variables as frequency (%). The McNemar test was used to identify statistically significant differences in categorical data for T0 and T1. Analyses were performed using SAS software version 9.4 (SAS Institute, Inc., Cary, NC, USA).

## 3. Results

Of 1800 physician staff members at ALGH, 481 physicians (26.7%) enrolled in the study at T0. Of these, 458 (95%) remained in the study at T1. A total of 458 participants completed the questionnaires at T0, and 452 participants completed the questionnaires at T1. The questionnaire data from these participants were analyzed for the current study. Percentages for each answer were calculated from these numbers.

At T0, 239/458 (52.2%) study participants were female and 219/458 (47.8%) were male. The median age was 43 (range 25–79 years old). At T0, 63 (13.8%) of the participants worked in high-risk settings. Of these, 21 worked in the Emergency Department (ED), 30 in the COVID-19 Intensive Care Unit, and 22 in the non-COVID-19 Intensive Care Unit (ICU). The demographics of the study participants are shown in Table 1. It is important to note that there are missing responses in the following categories that do not add up to the complete 458 participants: “Physician role” = 455, “Primary Practice Location” = 454, “Race/Ethnicity” = 458, “Age” = 457. Upon further review, the explanation for this is not a tabulation error but rather the result of omitted responses from participants. The percentages listed are correct and representative of total responses to each question listed.

In total, 363 subjects reported no chronic medical conditions. A total of 45 (9.9%) reported hypertension, five (1.1%) reported coronary artery disease, 34 (7.5%) reported asthma, 10 (2.2%) reported diabetes, and 14 (3.1%) reported autoimmune diseases. Nine participants (2%) were taking immunosuppressant medications. A total of 102 (22.5%) participants reported that they had traveled internationally between December 2019 and March 2020, with one subject returning from China.

At T0, 21 (4.4%) of the participants tested positive for SARS-CoV-2 IgG serum antibodies. At T1, 39 (8.5%) of the participants tested positive for the SARS-CoV-2 IgG antibody. Due to the relatively small number of participants that developed these antibodies, there were no statistically significant differences detected between those that did and those that did not develop antibodies. The significance of these findings will be further assessed after the completion of our study using all four time points.

At time T0, there were 306 (67%) physicians that reported exposure to COVID-19 infected patients and 148 (33%) physicians reported that they were not exposed to COVID-19 patients.

At time T1, there were 278 (62%) physicians that reported that they were exposed to COVID-19 infected patients and 174 (38%) physicians reported that they were not exposed to COVID-19 patients.

At T0 and T1, 111 (24.2%) and 130 (28.4%) of participants were involved with high-risk aerosol-generating procedures, respectively. Participants involved with high-risk aerosol-generating procedures had a higher risk of reporting exposure to COVID-19 infected patients at T0 (OR = 4.464 CI = 2.522–8.459, *p* < 0.001) and at T1 (2.426 CI = 1.551–3.872, *p* < 0.001). This risk occurred regardless of whether participants reported involvement in care for COVID-19 patients or persons under investigation for COVID-19 infection (PUI). Participants working in high-risk settings had a higher risk of reporting exposure to COVID-19 infected patients at T0 (OR = 6.919 (CI = 3.168–18.218, *p* = 1.036 × 10^−5^)).

At T0, 251 (55.4%) participants expressed moderate to extreme concern about becoming infected with COVID-19 (See Figure 1), and 221 (49.0%) noted a moderate to extreme concern expressed about the risk of their household members becoming infected with COVID-19 due to the participant’s potential of workplace exposure (see Figure 2).

Overall, participants were highly adherent to the use of personal protective equipment during patient care (see Table 2). The most used PPE at T0 and T1 were gloves (95%), gowns (89%, 88%), N95 masks (82.6%, 85.4%), face shield (72%, 67%) and surgical scrubs (59.3%, 58%).

In addition, the participants were highly adherent to social distancing practices (see Table 3). The most reported social distancing practices at T0 and T1 included wearing masks in public places (86%, 82.1%), avoiding mass gatherings or crowded places (85.1%, 85.6%), staying at least 6 feet from other people (83.8%, 83.4%), avoiding public transportation (78%, 83.8%, and not participating in small group gatherings (65.1%, 77.4%).

Based on the McNemar test, significant changes in subject behaviors or compliance between T0 and T1 were determined by comparing the proportions of subjects whose answers changed (either from yes-to-no or from no-to-yes). Several changes were noted, and odds ratios were calculated to assess for clinical significance with an example of the calculation displayed in Table 4. Table 4 demonstrates that subjects who had contact with COVID-19 infected patients were 2.56 times more likely to have had subsequent contact with COVID-19 infected patients relative to those who did not have contact with COVID-19 infected patients. These included having contact with COVID-19 infected patients, having not participated in group gatherings, having not attended a crowded place or mass gathering, avoiding the use of public transportation, ridesharing or taxis, using mail orders and delivery services for groceries, and changing one’s level of concern about coming into close contact with a COVID-19 patient as a healthcare provider (see Figure 3). The largest changes noted in PPE practices between T0 and T1 were the abandonment of cloth masks as PPE and a marked decrease in subjects that were not using PPE in the hospital (see Figure 4).

Specifically, in answer to the question about having contact with COVID-19 infected patients at work, 15.1% of subjects changed their answer from Yes to No while 10% of subjects changed their answer from No to Yes (*p* = 0.0343). For compliance with social distancing guidelines by not participating in group gatherings, 8.2% of subjects who reported not having attended group gatherings at T0 attended group gatherings at T1, while 20.2% of the subjects who attended group gatherings at T0 avoided group gatherings at T1, (*p* < 0.0001). Similar trends were observed for other social distancing measures. For example, significant shifts in social distancing attitudes towards public transportation between two time points were demonstrated. In total, 7% of subjects who avoided using public transport at T0 used public transportation at T1, while 13% of the subjects who used public transportation at T0 reported avoiding public transportation at T1 (*p* = 0.005). Uses of delivery service for groceries remained stable for the two time periods. Furthermore, 5.4% of subjects using food delivery services at T0 discontinued these services at T1, while 6.2% of subjects not using food delivery services at T0 changed to using food delivery services at T1.

## 4. Discussion

Physicians and other healthcare providers have been heavily impacted by the COVID-19 pandemic. With close personal exposure to a highly infectious and potentially deadly infection, several studies have revealed enhanced health and psychological burdens on healthcare providers in general [6,23,24]. As a group, physicians face specific risks due to frequent proximity to infected patients and the performance of procedures associated with exposure to infectious secretions [7,25,26]. Among physicians, sites of the highest exposure to viral burden include the emergency department (ED) [27], operating rooms [28] and intensive care units (ICU) [29]. The additional risk of COVID-19 exposure would also likely occur in healthcare workers in the COVID-19 intensive care unit. Physicians performing procedures such as intubation, bronchoscopy and care of patients during a cardiorespiratory arrest are known to be exposed to high viral burdens [30,31].

At present, the medical literature on the effect of COVID-19 on healthcare provider behavior has focused on several areas. These have included provider knowledge about the transmissibility of the virus, concerns about personal exposure to COVID-19 and the use of personal protective equipment [32,33,34]. These studies have not provided information about the concerns for the risk of acquisition of COVID-19 among physicians and the potential for spread to their families. Understanding the degree of concern experienced by physicians that are on the front lines of care for patients infected with COVID-19 is essential as the pandemic was demonstrated to have a significant impact on the physical and mental wellbeing of front-line workers. Furthermore, the actual use of PPE by physicians representing a wide variety of specialties and risk for COVID-19 exposure has not been previously investigated, while the individual social distancing practices of physicians have not been fully elucidated. Prior studies have not used the method of longitudinal review of these behaviors. This is an important area of study, as the level of COVID-19 infections in individual communities has evolved over time. The effects of fluctuating levels of infections within communities on the concern for physician acquisition of infection or their risk of spreading it to their families and communities are currently unknown. The longitudinal nature of the pandemic and its effect on PPE usage and social distancing practices was also not previously investigated. Our study was performed to develop a better understanding of these factors among physicians working in a community hospital setting.

The IgG antibody test developed by ACL Laboratories used in this study utilizes the Abbott Architect platform that detects the N protein of SARS-COV-2 IgG. The Abbott Architect SARS-CoV-2 IgG assay is a qualitative test for IgG antibodies to the nucleocapsid protein of SARS-CoV-2 in serum and plasma. The test is used from patients who are suspected of past infections with COVID-19, although positive results can occur in patients with past or present COVID-19 infections [35]. Nonreactive results may not rule out current COVID-19 infection if they are measured shortly after exposure. A positive result in the current study was used to validate the assumption that the person had developed an adaptive immune response to the virus. This occurs after having been exposed and having recovered from COVID-19 virus infection.

The antibody positivity rates found among physicians in this study are similar or lower than those found in previous studies performed in the United States. For example, several groups of physicians were tested for COVID-19 IgG studies from New York City, the epicenter of COVID-19 infections in the early portion of the pandemic. These included anesthesiologists, intensivists, surgeons, general practitioners, and resident physicians [36,37,38,39,40]. Further studies to document COVID-19 exposure in healthcare providers using the measurement of IgG were performed worldwide, including Italy and the United Kingdom [41,42]. A prior published study utilizing the same qualitative test for IgG antibodies to the nucleocapsid protein of SARS-CoV-2 in serum and plasma was performed in our hospital system. In total, 14,921 adults employed through our healthcare system underwent COVID-19 IgG testing between 8 June and 10 July 2020 [17]. Of these, 535 (3.59%) were positive for COVID-19 IgG. In the present study, tests for COVID-19 were performed in physicians only and testing took place in September and December of 2020. Despite being a more selected group of healthcare practitioners (attending physicians and residents), and testing being performed two and five months later, COVID-19 IgG positivity rates in our study group of 4.4% in September 2020 closely matched the positivity rate of all employees being tested in July 2020. The increase in IgG positivity rate of 8.5% in December likely reflects the marked increases of active COVID-19 seen in the State of Illinois in the Fall and early Winter of 2020 [43]. Because of our relatively small number of IgG-positive subjects, we were unable to study specific risks for COVID-19 infections in our study group. A previous study of the acquisition of COVID-19 infection in healthcare workers showed that between March 2020 and June 2020, healthcare workers working in patient care areas were four times more likely to become infected with COVID-19 than those not working in patient care areas [44].

Our study represented a cross-section of physicians in our hospital, including approximately 76% attending physicians and 14% residents and fellows. Furthermore, there was a variety of work-related degrees of exposure to COVID-19 infected patients, activities in high-risk settings for exposure and involvement in high-risk procedures among the participants. The design of our study and the prospective nature of data collection allowed for an analysis of the stability of these practices. This represents a unique aspect of our study in comparison to other published literature during the pandemic.

Despite the diversity of these factors, exposure to patients with suspected or confirmed COVID-19 infection was consistent for the three-month interval between collection periods, with 67% and 62% reporting exposure at T0 and T1, respectively. In addition, steady behaviors were seen within the study group. Participants reported a consistently high degree of compliance with PPE use for the three-month interval between collection periods, with all participants reporting some form of PPE use at both intervals.

A high degree of compliance with social distancing measures also was noted during the three-month interval between collection of COVID-19 IgG levels and work and lifestyle-related questionnaires. For example, more than 85% of study participants avoided crowded places or mass gatherings at T0 and T1. The motivation for strict adherence to social distancing measures in our study group could in part reflect their concern about the risk of infecting other persons in the community because of their potential exposure to COVID-19 infected patients at work.

The design of our study and the prospective nature of data collection allowed for an analysis of the stability of these practices. This represents a unique aspect of our study in comparison to other published literature during the pandemic. Some concern regarding ongoing adherence to social distancing practices has resulted in resistance to these practices in some populations. Overall fatigue with the rigors of social distancing resulted in protests related to lockdown imposed on European populations during the second wave of infection beginning in September 2020 [45,46]. A high level of belief in social distancing and trust in both public health agencies and ruling political systems was associated with compliance with these practices during the COVID-19 pandemic [47].

We used the Two-Sample McNemar test [48] to evaluate for statistically significant changes in social distancing practices within the data collected in September and December of 2020. Because we did not have a preconceived understanding of how changes in these factors might occur, nor a full awareness of the advent of the large increase in COVID-19 cases in our immediate geographic area or in the State of Illinois, the post-collection statistical analysis took place for the purpose of hypothesis testing. This form of analysis was used to compare differences in dichotomous items during an interval between the two data collection periods. This analysis revealed an increase in reported exposure to COVID-19 infected patients during the time interval. The largest changes in social distancing practices found included 8.2% of subjects who reported not having attended group gatherings at T0 attended group gatherings at T1 while 20.2% of the subjects who attended group gatherings at T0 avoided group gatherings at T1, (*p* < 0.0001). Additionally, a statistically significant change in increased avoidance of public transportation was noted during the three-month time interval. More formal investigations of these changes in social distancing during a pandemic have the potential to guide public policy in the future.

This study is an initial evaluation of a one-year longitudinal study. The study involved the development of a new survey to review PPE use, physician concern regarding their risk of developing and spreading COVID-19 infection and social distancing practices over time. This survey and the information obtained is an opportunity to evaluate changes in these practices over time. In future analyses, these data will be utilized for model building and for assessing how the risks of exposure affect this behavior.

Since this study was initiated prior to the introduction of vaccines against COVID-19 infection, our initial analysis does not include data related to changes in attitudes and practices following vaccination. It is anticipated that once study subjects were vaccinated, their level of concern about their risk for acquiring or spreading COVID-19 infections will have diminished. Likewise, the study could not have anticipated the importance of vaccine acceptance, an important issue in the global pandemic battle. Although public policies including the use of a “green pass” have shown to have complex effects on vaccine acceptance [49], our subject group displayed a high vaccination rate once the vaccine became available.

In summary, our initial analysis of data collected during the first wave and in advance of the second wave of the COVID-19 pandemic demonstrates that physicians in a large community practice stayed relatively protected from exposure to COVID-19, based on the measurement of serum IgG antibodies. This group of physicians, with a variety of degrees of risk to COVID-19 infected patients, expressed considerable concern about the hazard of acquiring COVID-19 infection and their risk of exposing household and community members to the disease because of their risk of workplace exposure. In keeping with this finding, they consistently adhered to PPE practices. As the pandemic progressed, certain social distancing practices including the avoidance of crowds and public transportation increased in the study group.

## Figures and Tables

**Figure 1 healthcare-10-00285-f001:**
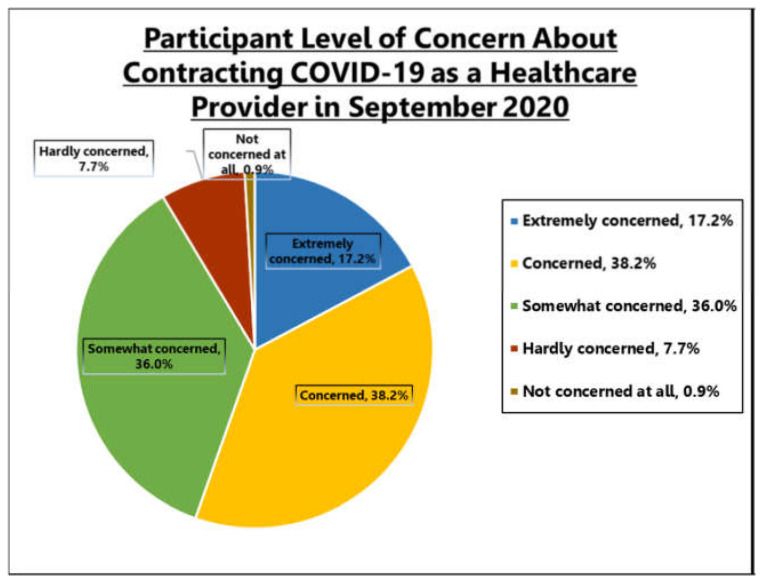
Level of concern of contracting COVID-19 as Healthcare Worker.

**Figure 2 healthcare-10-00285-f002:**
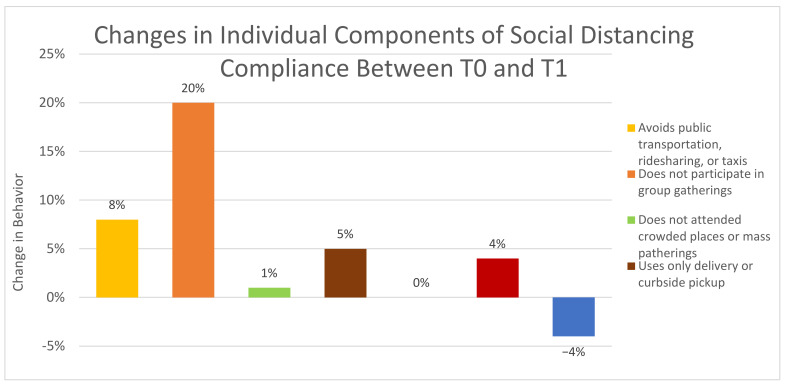
Household concern of contracting COVID-19.

**Figure 3 healthcare-10-00285-f003:**
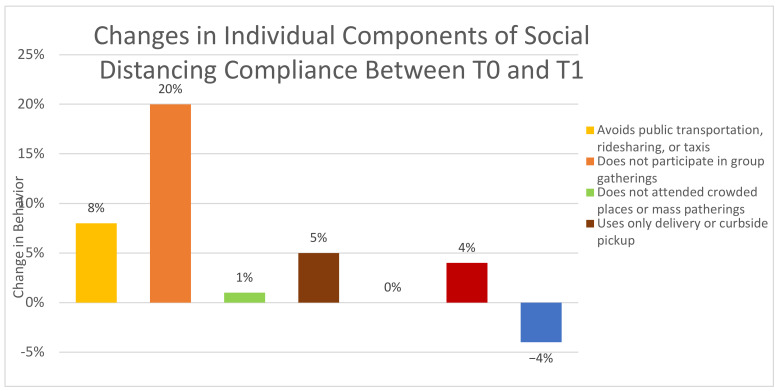
Changes in individual components of social distancing compliance between September 2020 (T0) and December 2020 (T1).

**Figure 4 healthcare-10-00285-f004:**
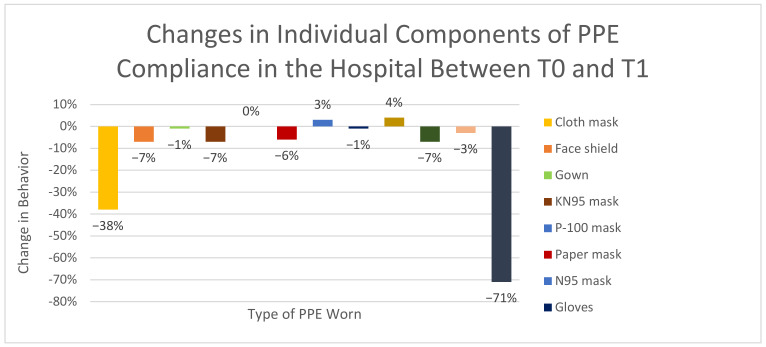
Changes in individual components of PPE compliance in the hospital between T0 and T1.

**Table 1 healthcare-10-00285-t001:** Demographics of the study population category.

Sub-Category	Frequency (*n*)	Percent (%)
Gender		
Male	219	47.8
Female	239	52.2
Age (years)		
25–39	190	41.6
40–59	195	42.7
60–79	72	15.8
Race/Ethnicity		
White	323	70.5
Asian/Pacific Islander	97	21.2
Hispanic/Latino	6	1.3
Black/African American	4	0.9
Other	28	6.1
Physician Role		
AMG Attending	197	43.3
Non-AMG Attending	150	33.0
Resident	88	19.3
Fellow	20	4.4
Primary Practicing Location		
Outpatient Ambulatory	178	39.2
COVID-19 Medical/Surgical Floor	34	7.5
Non-COVID-19 Medical/Surgical Floor	179	39.4
Emergency Department	21	4.6
COVID-19 Intensive Care Unit	20	4.4
Non-COVID-19 Intensive Care Unit	22	4.8

Participant Demographics, Roles within Advocate Lutheran General Hospital (ALGH)/Advocate Medical Group (AMG), and Primary Locations of Employment. Reported at T0 in September 2020.

**Table 2 healthcare-10-00285-t002:** Reported use figure 2020. PPE.

	September 2020 (T0)	December 2020 (T1)
Gloves	95.2% (*n* = 432)	95.1% (*n* = 430)
Gown	89.0% (*n* = 404)	88.1% (*n* = 398)
Scrubs	59.3% (*n* = 269	58.0% (*n* = 262)
Goggles	50.0% (*n* = 227)	52.0% (*n* = 235)
Face Shield	72.0% (*n* = 327)	67.0% (*n* = 303)
N95 Mask	82.6% (*n* = 375)	85.4% (*n* = 386)
KN95 Mask	5.9% (*n* = 27)	5.5% (*n* = 25)
P-100 Mask	1.1% (*n* = 5)	1.1% (*n* = 5)
Paper Mask	23.8% (*n* = 108)	22.8% (*n* = 103)
Cloth Mask	4.6% (*n* = 21)	2.9% (*n* = 13)
Powered Air-Purifying Respirator	6.6% (*n* = 30)	6.2% (*n* = 28)
No PPE	1.5% (*n* = 7)	0.4% (*n* = 2)

**Table 3 healthcare-10-00285-t003:** Study participant responses to questions regarding social distancing practices in September and December of 2020.

Social Distancing Measure	September 2020 (T0)	December 2020 (T1)
I stay at least 6 feet away from other people	83.8% (*n* = 377)	83.4% (*n* = 377)
I do not participate in group gatherings	65.1% (*n* = 293)	77.4% (*n* = 350)
I have not attended a crowded place or a mass gathering	85.1% (*n* = 383)	85.6% (*n* = 387)
I avoid using public transportation, ridesharing or taxis	78.0% (*n* = 351)	83.8% (*n* = 379)
I use mail orders and delivery services for groceries	20.9% (*n* = 94)	21.7% (*n* = 98)
I wear a mask in public spaces	86.0% (*n* = 387)	82.1% (*n* = 371)
I only purchase groceries and other products by delivery or curbside pickup	9.3% (*n* = 42)	9.7% (*n* = 44)
I do not follow any social distancing measures	0% (*n* = 0)	0% (*n* = 0)

**Table 4 healthcare-10-00285-t004:** Example calculation of odds ratio and relative risk for the consistency of COVID-19 contact by providers from T0 to T1. Subjects who had contact with COVID-19 infected patients were 2.56 times more likely to have had subsequent contact with COVID-19 infected patients relative to those who did not have contact with COVID-19 infected patients.

Odds Ratio and Relative Risk of COVID-19 Contact Consistency			
Statistic	Value	95% Confidence Limits	
Relative Risk	2.557	1.977	3.3071

## Data Availability

Deidentified data will be made available to interested parties. Formal requests for data will be reviewed by the study authors. Only requests that are approved by the authors will receive the study data.

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
