# Peer review of "A Prospective, Longitudinal Evaluation of SARS-CoV-2 COVID-19 Exposure, Use of Protective Equipment and Social Distancing in a Group of Community Physicians"

_healthcare, 2022, doi:10.3390/healthcare10020285_

Round 1

Reviewer 1 Report

All staff physicians at Advocate Lutheran General Hospital were invited to participate. COVID-19 IgG antibody was measured and an online questionnaire was completed. The questionnaire assessed the risk of COVID-19 exposure, PPE usage, concern for contracting COVID-19, the performance of high-risk procedures, work in high-risk settings, and social distancing practices. Testing was performed in September (T0), and December 2020 (T1), at the height of the global pandemic Results 481 (26.7%) of 1800 AGLH were enrolled at T0, and 457 (95%) at T1. 21 (4.3%) and 39 (8.5%) participants had antibodies at TO and T1. 63 (13.8%) worked in high-risk settings, and 111 (24.2 %) performed high-risk procedures. Participants working in high-risk settings had increased exposure to COVID-19 infected patients, (OR= 4.464 (CI=2.522-8.459, p<0.001). Participants were highly adherent to the use of PPE and social distancing practices including mask-wearing in public (86%, 82.1%), avoiding crowds (85.1%, 85.6%), six feet distancing (83.8%, 83.4%), and avoiding public transportation (78%, 83.8%). 251 (55.4%) participants expressed moderate to extreme concern about becoming infected with COVID-19. Conclusions and Relevance Among a group of community physicians, consistent PPE use, and social distancing practices were common. These consistent practices were associated with a low level of initial acquisition of COVID-19 infections, and a relatively low longitudinal risk of infection.

In General: it's a good paper and the subject of the manuscript is applicable and useful. 

Title: the title properly explain the purpose and objective of the article

Abstract: abstract contains an appropriate summary for the article, language used in the abstract is easy to read and understand, there are no suggestions for improvement.

Introduction: authors do provide adequate background on the topic and reason for this article and describe what the authors hoped to achieve.

Results: the results are presented clearly, the authors provide accurate research results, there is sufficient evidence for each result.

Conclusion: in general: Good and the research provides sample data for the authors to make their conclusion.

Grammar: Need Some revision.  

Finally, this was an appealing article, in its current state it adds much new insightful information to the field. Therefore, I accept that paper to be published in your journal

Author Response

Recommendations from reviewer: "Grammar: Need Some revision."

Response to Reviewer:

Thank you for your observation and taking the time to read out submitted manuscript. We have re-read the manuscript several times and listed the below changes to grammatical errors:

Line 47, sentence updated for improved grammar and clarity and now reads "Physicians involved in the performance of high-risk procedures (such as intubation and bronchoscopy) are likely at higher risk for direct exposure to the virus (7)."
Line 70, removed "and occurrences of infection."
Line 198, removed "a" before "COVID-19 patients"
Line 224, parenthesis closed with ")"
Line 228, "providers" changed to "provider"
Line 279, excessive spaces present, deleted.
Line 385, improper bolding of one letter, changed.
Lines 392 and 393, the "." after Dr was missing on two separate occasions, this was added.

Reviewer 2 Report

The numbers are not sufficiently explained and are not consistent with the corresponding numbers in Table 1, also comparing the results in the abstract and in the main part (e.g. T1: 457 participants in abstract, but 458 in main part). Other examples: 63 (worked in high-risk settings) and 111 (performed high-risk procedures) refer obviously to T1 with 457 participants in the abstract, but in the main part to T0, which would mean 13.1% for 63; there are only 454 participants with primary practicing location in Table 1, and so on and so on. Please, check your numbers over the whole paper.

Table 2 have to be extended by the corresponding numbers for “mask wearing in public”, “avoiding crowds”, “six feet distancing”, and ”avoiding public transportation”.

A presentation of a 2x2 table (at least for one example) for the calculation of the odds ratios, corresponding confidence intervals and p-values for the McNemar test, would be very helpful for an easy understandable reading of the paper.  

Author Response

Reviewer Point #1:

The numbers are not sufficiently explained and are not consistent with the corresponding numbers in Table 1, also comparing the results in the abstract and in the main part (e.g. T1: 457 participants in abstract, but 458 in main part). Other examples: 63 (worked in high-risk settings) and 111 (performed high-risk procedures) refer obviously to T1 with 457 participants in the abstract, but in the main part to T0, which would mean 13.1% for 63; there are only 454 participants with primary practicing location in Table 1, and so on and so on. Please, check your numbers over the whole paper.

Response Point #1:

Thank you for pointing out these discrepancies. The number of participants was in fact 458 at T0 and the abstract has been corrected. In regard to the math of the percentage question, with 63 participants out of 458, that number would be 13.8% as listed. You are correct in identifying that there were four responses from participants missing in the "Primary Practicing Location.” Additionally, there are other missing responses in the following categories that do not add up to the complete 458 participants: “Physician role” = 455, “Location” = 454, “Race” = 458, “Age” = 457. Upon further review, the explanation for this is not a tabulation error but rather the result of omitted responses from participants. We have updated our results section to explain this discrepancy with the explanation beginning at line 165. The percentages are correct as listed based on these numbers.

Regarding checking the numbers over the whole paper, we also realized that Figure 2 seems to have been omitted from the submission and this was added to the submitted copy.

The remainder of the data listed is correct when compared to our source data.

Reviewer Point #2:

Table 2 have to be extended by the corresponding numbers for “mask wearing in public”, “avoiding crowds”, “six feet distancing”, and ”avoiding public transportation”.

Response Point #2:

Table 3 has the data and completely reports the requested survey questions, including “mask wearing in public”, “avoiding crowds”, “six feet distancing”, and ”avoiding public transportation.” Upon further review, it appears Table 3 was not included in the formatted submission after we uploaded it to the website and was accidentally omitted from the copy that this reviewer read. The attached version shows Table 3 with this data.

Reviewer Point #3:

A presentation of a 2x2 table (at least for one example) for the calculation of the odds ratios, corresponding confidence intervals and p-values for the McNemar test, would be very helpful for an easy understandable reading of the paper. 

Response Point #3:

The following line was added to our Methods section (Line 148) for completeness: “For example, consistency of exposure and use of high-risk aerosolizing procedures were compared from To to T1 and assessed for statistical significance.”

We then added a table (Table 4, Line 245) that demonstrates the calculation of the odds ratio and relative risk with associated confidence intervals. The is also referenced in the body of the text starting at line 231 and reads, “Several changes were noted, and odds ratios were calculated to assess for clinical significance with an example of calculation displayed in Table 4. Table 4 demonstrates that subjects who had contact with COVID-19 infected patients were 2.56 times more likely to have had subsequent contact with COVID-19 infected patients relative to those who did not have contact with COVID19 infected patients.